# Molecular Regulation of the Response of Brain Pericytes to Hypoxia

**DOI:** 10.3390/ijms24065671

**Published:** 2023-03-16

**Authors:** Robert Carlsson, Andreas Enström, Gesine Paul

**Affiliations:** 1Translational Neurology Group, Department of Clinical Science, Wallenberg Neuroscience Centre and Wallenberg Centre for Molecular Medicine, Lund University, 22184 Lund, Sweden; 2Department of Neurology, Scania University Hospital, 22185 Lund, Sweden

**Keywords:** pericyte, hypoxia, stroke, brain, vasculature, regulator of G-protein signaling 5, transcription factors

## Abstract

The brain needs sufficient oxygen in order to function normally. This is achieved by a large vascular capillary network ensuring that oxygen supply meets the changing demand of the brain tissue, especially in situations of hypoxia. Brain capillaries are formed by endothelial cells and perivascular pericytes, whereby pericytes in the brain have a particularly high 1:1 ratio to endothelial cells. Pericytes not only have a key location at the blood/brain interface, they also have multiple functions, for example, they maintain blood–brain barrier integrity, play an important role in angiogenesis and have large secretory abilities. This review is specifically focused on both the cellular and the molecular responses of brain pericytes to hypoxia. We discuss the immediate early molecular responses in pericytes, highlighting four transcription factors involved in regulating the majority of transcripts that change between hypoxic and normoxic pericytes and their potential functions. Whilst many hypoxic responses are controlled by hypoxia-inducible factors (HIF), we specifically focus on the role and functional implications of the regulator of G-protein signaling 5 (RGS5) in pericytes, a hypoxia-sensing protein that is regulated independently of HIF. Finally, we describe potential molecular targets of RGS5 in pericytes. These molecular events together contribute to the pericyte response to hypoxia, regulating survival, metabolism, inflammation and induction of angiogenesis.

## 1. Brain Capillaries Are Crucial for Oxygen Delivery

Oxygen is fundamental for the survival and for the respiratory chain metabolism of the human body. As a result, an adequate adaptive response to changes in oxygen levels is needed to maintain vascularization and oxygen homeostasis in the organism, especially under conditions of hypoxia. Oxygen levels may change due to an increased demand of the tissue or due to fluctuations in oxygen supply. Those changes can normally be compensated for by increased lung activity, oxygen uptake and increasing the blood circulation to the tissues. In pathology, however, those compensatory mechanisms are not sufficient.

The adaption to fluctuations in oxygen is associated with particular acute or chronic cellular responses. Those responses require specific molecular mechanisms that allow the cells to sense the changes in oxygen and to link them to transcriptional or post-transcriptional molecular responses. The molecular events then enable the organism to make changes corresponding to oxygen availability or to its metabolic needs in a fast and adequate manner. 

Such adaption to oxygen supply or demand is especially important in the brain. The human brain consumes ca 20% of the total available oxygen at rest. As a result, the brain vasculature has adapted to meet this high oxygen and energy demand by forming a 650 km long capillary network supplying the brain with oxygen and nutrients [1]. The gas exchange is taking place at the capillaries that form the smallest entity of this vast vascular network and constitute the interface between the blood and the brain. Here, neurons are located in very close connection with perivascular microglia and capillaries, forming the neurovascular niche (Figure 1). 

Endothelial cells and perivascular pericytes embedded in the basal membrane comprise the microcapillary wall and, together with astrocytic end-feet, form a highly specialized and tightly regulated border between the blood circulation and the brain parenchyma: the blood–brain barrier (BBB) [2]. 

## 2. Hypoxia-Induced Angiogenesis 

One of the default responses of tissues to hypoxia is the initiation of angiogenesis. 

Angiogenesis is the formation of new blood vessels from the pre-existing vessel network, with the purpose to increase blood circulation and oxygen supply. This is a process initially allowing the rapidly growing embryo to adapt its vascular system. In the adult, angiogenesis occurs usually in a pathological context in response to hypoxia or inflammation [3]. 

The initiation of angiogenesis starts with the induction of a tip endothelial cell by angiogenic factors secreted by neighboring cells and triggered by the hypoxic environment. This is followed by digestion of the basement membrane and the detachment of pericytes from the blood vessels, which allows the formation of a new vessel sprout by endothelial cells. This sprout elongates to form a new, yet immature, blood vessel. When two tip endothelial cells meet, pericytes are recruited through communication with endothelial cells and a new basement membrane is formed, resulting in maturation of the newly formed blood vessel [4,5]. Importantly, tip endothelial cells secrete platelet-derived growth factor-BB (PDGF-BB), which is recognized with high affinity by PDGF receptor β (PDGFRβ)-expressing pericytes. This leads to pericyte recruitment, which then stabilizes the blood vessel [6] (Figure 2).

## 3. Brain Hypoxia and Pathology

Acute and prolonged mismatch between metabolic supply and demand has severe consequences for the human brain. Thus, impaired oxygen delivery to the tissue and functional impairment of the neurovascular unit are characteristics of several cerebrovascular and neurodegenerative diseases. The condition most widely studied and connected with acute hypoxia is ischemic stroke, which is the second leading cause of death worldwide and one of the main reasons behind adult disability [7]. 

In ischemic stroke, the lack of oxygen and glucose to the affected brain regions leads to a cascade of molecular and cellular events that arises within seconds to minutes from the ischemic insult and progresses for several weeks [8]. The few hours following the initial hypoxic event correspond to the hyperacute phase of stroke, which is characterized by cell death, microglial and astrocytic activation and breakdown of the BBB. This leads to an increase in BBB permeability and vascular leakage, with this being deleterious to the neighboring cells [9]. During the chronic phase after stroke, starting days after the injury, endogenous repair mechanisms take place that include vascular remodeling, neural plasticity and scar formation [10,11].

The human brain has not evolved to be adequately prepared for such a sudden and extensive hypoxia as an ischemic stroke. Nonetheless, apart from increasing collateral blood flow, the adult brain seems to react with an attempt for angiogenesis. Angiogenesis, however, may not be beneficial due to its timing in the context of a pathological cascade of events in ischemic stroke. Angiogenesis is a double-edged sword as newly formed vessels are immature and can result in BBB leakage, especially when pericyte recruitment is impaired. This will lead to oedema and worsen the stroke outcome [9,12,13].

Therefore, the cellular response to such an extensive and prolonged hypoxia may become dysregulated and result in negative consequences. Better knowledge of the cellular and molecular response to hypoxia is crucial to identify targets that can be modulated to achieve a more balanced adaption to the pathological state. 

## 4. Pathophysiological Response of Pericytes to Hypoxic Conditions

Pericytes are one of the first responders to cerebral hypoxia [14,15]. They line the entire microvasculature in the brain and have a unique position at the abluminal site of the capillaries at the blood–brain interface. While ubiquitously present in all organs, the pericyte density is highest in the central nervous system (1:1 to 1:3 pericyte to endothelial cell ratio), as compared to, for example, skeletal muscle (1:100 ratio) [16,17]. During embryonic vascular development, pericytes stabilize and maintain vessels [18,19], whereas in the adult brain, they maintain BBB integrity, have secretory abilities and likely function in inflammation [20,21] and play an important role in angiogenesis under pathological conditions [22,23,24,25,26,27]. 

When activated in response to pathological stimuli such as hypoxia, pericytes undergo specific modifications in morphology, involving changing from a flat cell soma with longitudinal and thin processes [28] to a more bulging cell soma with shorter processes [28,29]. This pattern is predominantly seen following injury and during the early stages of angiogenesis and is often associated with expression of typical markers [30,31]. 

## 5. Transcriptional Responses to Hypoxia in Pericytes

Transcriptional responses that play a role in the adaptation to the hypoxic environment are important. The transcriptional response produces proteins that promote survival of pericytes or other cell types in low oxygen concentrations by changing their metabolism to be less dependent on oxidative phosphorylation, the citric acid cycle or glycolysis in hypoxia [32]. The metabolic changes that confer cell survival in hypoxia appear to vary between cell types, e.g., endothelial cells and astrocytes seem to be more resilient to hypoxic insults than other cell types, such as neurons [33]. 

We have previously studied the transcriptomic changes in pericytes in hypoxia and postulated that, in particular, four transcription factors are involved in regulating the majority of transcripts that change between hypoxic and normoxic pericytes [34]. The transcriptomic changes to hypoxia are rapid and may, in some cases, even involve direct changes in transcription factor function. Transcription factor translocation to the mitochondria and activity in the mitochondrial electron transport chain prior to any hypoxia-induced DNA binding precedes transcriptional activation in the cellular nucleus [35,36]. 

In the following part, we describe some of these acute transcriptional responses in pericytes exposed to hypoxia, illustrating tentative mechanisms that are necessary for the survival of pericytes during the first hours after hypoxia onset. 

### 5.1. NFκB

Nuclear factor of kappa light polypeptide gene enhancer in B-cells (NFκB) is a transcription factor expressed in many tissues and in pericytes. NFκB is a homo/hetero-dimeric protein that can be assembled out of five structurally different variants: NF-κB1 (p50), NF-κB2 (p52), RELA (p65), RELB and/or c-REL, respectively [37,38]. The most transcriptionally active heterodimer is RELA/RELB. RELA/RELB binds to the kappa B consensus DNA sequence in DNA regulatory elements and activates the transcription of adjacent target genes. Without activation signals, NFκB is kept quiescent—in a bystander state, where NFκB is sequestered in the cytoplasm by binding to the inhibitor of NFκB (Iκb) (Figure 3). Iκb is also expressed throughout the tissues of the mammalian organism, but can respond to different stimuli, e.g., metabolic stress (H2O2, hypoxia) through to full activity in human osteosarcoma and breast cancer cell lines, in a matter of 5 to 15 min [39]. If Iκb is serine phosphorylated by such stimuli, the inhibitory binding to NFκB is lost, and NFκB migrates to the nucleus and regulates transcription of target genes [40] (Figure 3). Hypoxia in micro-vessels from hypoxic rat brains (1 h 6% O2) results in activation and recruitment of NFκB, HIFs and other transcription factors to their target DNA binding sites in a hypoxia-dependent fashion [41]. This suggests that the transcriptional niche induced by hypoxia involves a multitude of active transcription factors, possibly fine-tuning the response to hypoxia depending on the cell type’s location and requirements. When it comes to NFκB, the canonical activation pathway is required for robust and early DNA binding activity of NFκB and IκBα phosphorylation in cancer cells [39]. The hypoxia-induced DNA binding activity of NFκB here induced the transcription of IL8, a cytokine that is involved in regulation of inflammation and angiogenesis and induced by hypoxia in pericytes [34]. SiRNA p65 silencing experiments in HeLa cells show that COX2 is regulated by hypoxic activation of NFκB, and to some extent, by HIF1α [42]. NFκB transcriptional activity reporters are activated in accordance with the above findings in in vivo hypoxia assays [42], showing that NFκK is important for the response to hypoxia in vivo.

Adenosine is rapidly released in the hypoxic brain; according to micro-dialysis experiments, this occurs in a matter of 30 s to five minutes [43,44,45,46]. In astrocyte mono-cultures, IL6 transcription is dependent on adenosine activation of NFκB [47]. IL6 activates STAT3, and NFκB is likely an important factor initiating this cascade of cellular/pericyte adaptation to an hypoxic insult (see Section 5.3). The IL6 transcript is upregulated in pericytes as early as 2 h after hypoxic treatment [34].

The exact genes regulated by hypoxic activation of NFκB likely differ from cell type to cell type. NFκB have been suggested to bind to, and regulate, genes induced by both oxygen and oxygen glucose deprivation (OGD) in pericytes [34]. A NFκB-induced transcriptional program, dependent on Iκb-phosphorylation and subsequent recruitment of NFκB to available DNA binding sites in gene regulatory regions, is then then rapidly established in hypoxic pericytes [34]. The hypoxia/NFκB-induced proteins likely help pericytes to cope with the initial metabolic stress of oxygen deprivation to readapt and survive long-term oxygen deprivation. Gene set enrichment analysis (GSEA) of pericytes treated with 2 h of hypoxia or OGD compared to normoxia shows that NFKB is the top pathway regulated by this treatment, suggesting that other highly ranked GSEA hallmark metabolic pathways, such as glycolysis, are, at least in part, regulated by NFκB [34].

### 5.2. C-JUN/C-FOS

Jun proto-oncogene (C-JUN) is an immediate early transcription factor. C-JUN forms the AP1 transcriptional activation complex together with the proto-oncogene transcription factor C-FOS. The canonical C-JUN activation route is via the kinase activity of the c-Jun N-terminal kinase (JNK), where C-JUN is phosphorylated on two residues [48] (Figure 3). The dually phosphorylated C-JUN is then capable of dimerization with C-FOS, which leads to nuclear localization and DNA binding capability of the AP-1 heterodimer. DNA-bound AP-1 can transactivate adjacent genes to regulate pro-apoptotic/anti-apoptotic, mitogenic, cell-growth or differentiation signals, among others. In the field of hypoxia, gene regulation, sensing of oxygen and subsequent activation of proteins in an oxygen-dependent manner is a prerequisite to be considered as an inducer of hypoxic gene regulatory responses. For C-JUN, the oxygen-sensing mechanisms are yet to be unraveled and might involve loss of function in the respiratory chain. The sensing of hypoxia by the mitochondrial respiratory chain produces the reactive oxygen species (ROS) superoxide through complex III [49], which induces the activation of C-JUN [50]. C-JUN has been described to be activated by hypoxia-induced ROS production via RAC1 [51]. The molecules sensing hypoxia in this scenario would belong to the mitochondrial respiratory chain. There are, however, alternative possibilities. In hypoxia, ROS production potentially modifies DNA integrity and induces single-strand and double-strand breaks. It has been suggested that hypoxia does not induce DNA damage per se, and that the observed DNA damage in hypoxia arises from hypoxic inhibition of DNA repair enzymes (which would not be needed if DNA damage was absent) [52]. As an example of C-JUN activation via DNA damage, DNA damage can, on its own, induce the activation of JNK kinase activity and activation of C-JUN and/or sirtuin 6 (SIRT6) through phosphorylation [53]. Regardless, DNA damage via micro-irradiation induces very fast DNA repair mechanisms with a latency of only a few seconds in murine embryonic fibroblasts or neuroblastoma cells [54], relying on immediate phosphorylation events. The mitochondrial sensing of oxygen deprivation likely results in DNA damage by ROS production. DNA damage is, in turn, rapidly recognized by the DNA repair machinery activating C-JUN kinase JNK, and C-JUN transcriptional activation of hypoxia and DNA-damage-induced genes. C-JUN activates a multitude of target genes (IL8 etc.) as a response to, for example, hypoxia, metabolic stress or receptor ligand signaling [48]. Although the exact mechanism of hypoxic activation of C-JUN is unknown, the reactivity of the C-JUN pathway to hypoxia motivates further studies on its effects on hypoxic cells and stroke.

In pericytes, C-JUN activates some target genes as early as 2 h after hypoxic or OGD treatment and is the major contributor of early transcriptional changes to a number of genes regulated within 2 h of OGD [34]. At 6 h of hypoxia, HIF1α-associated activated genes outnumber the genes regulated by C-JUN. In OGD, the situation is reversed, with more genes regulated by C-JUN, indicating that glucose deprivation is contributing to C-JUN activity in synergy with hypoxia [34].

### 5.3. STAT3

Monomeric signal transducer and activator of transcription 3 (STAT3) proteins are confined to the cell cytoplasm and cannot exert DNA binding or transcriptional activity [55,56]. Janus activate kinases (JAKs) respond to interleukin 6 (IL6) or interleukin 11 (IL11) receptor activation by phosphorylating STAT3, respectively [57,58] (Figure 3). Phosphorylation of STAT3 results in its dimerization, either by homodimerization or heterodimerization to STAT1 [55]. Enhancer or promoter DNA binding of the dimerized STAT3 protein results in activation of gene transcription [55]. STAT3 has been shown to colocalize to the inner mitochondrial membrane upon pS727 phosphorylation [59] (Figure 3). This colocalization enhances the activity of the mitochondrial respiratory chain complexes I and II, as well as ATP synthase (complex V) [59].

The mechanism by which hypoxia induces STAT3 phosphorylation is unknown, but it has been speculated that NFκB can trigger IL6 production in response to hypoxia-dependent adenosine release [47]. This is not likely to account for the initial pSTAT3 induction after 5 min of hypoxia given that NFκB transcriptional activity and translation of the IL6 target gene probably requires a longer time of hypoxic exposure.

Crosstalk between STAT3 and NFκB has been suggested to occur after hypoxic stimulation [35], resulting in STAT3- and ROS-dependent translocation of RELA to mitochondria. For STAT3 and NFκB both, DNA binding does not seem to induce the hypoxia regulatory function, but instead direct translocation to the mitochondria.

We, and others, have shown that STAT3 is activated by hypoxia and OGD [34,60] (Figure 3). Indeed, the activation of tentative STAT3 target genes after 2 h of hypoxia or OGD precedes the activation of most tentative target genes of HIF1α [34]. In agreement, nucleus-localized STAT3 protein phosphorylation is increased already after 5 min of hypoxia or OGD, whereas full HIF1α stabilization takes 12 h [22,34]. Although it is not clear from our experiments that STAT3 mitochondrial association takes place and/or what function it has in hypoxia, we can observe hypoxia or OGD-induced pS727 STAT3 in pericytes after 5 min of treatment [34,55]. Rather than exerting transcriptional regulation, post-translational modifications, e.g., direct phosphorylation of pS727, induced by hypoxia seems to regulate initial transcription factor responses to hypoxia using protein moieties active in the mitochondria other than the DNA binding domain [36]. In transgenic mice overexpressing a DNA-binding-deficient STAT3 localized to the mitochondria, ROS production is lowered in the mitochondrial electron transport chain when compared to wild-type mice not expressing the STAT3 mutant [36]. This suggests that STAT3 has at least two functions in hypoxia: 1: localization and activity in the mitochondria, 2: transcriptional activation of STAT3-dependent target genes in the nucleus [34,60].

### 5.4. HIF1α

Hypoxia-inducible factor 1 alpha (HIF1α) is a basic helix loop helix (bHLH) transcription factor, and a prototypic hypoxia-stabilized protein regulated by prolyl hydroxylases (PHD1-3), that confers its degradation in the presence of oxygen [61,62,63,64] (Figure 3). Therefore, hydroxylation of the HIF1α protein silences the activity of HIF1α in two different ways, either by protein destabilization or inhibition of transcriptional activity [61,65].

The PHD-dependent hydroxylation of the newly synthesized HIF1α polypeptide leads to subsequent protein degradation through multiubiquitination and transport to the proteasome.

The primary oxygen sensor in the HIF pathway appears to be PHD2 [61]. Oxygen leads to PHD-dependent hydroxylation of two phosphorylated proline (P) amino acids (AAs) in the HIF1α protein (pP-402, pP-564). In addition, a phosphorylated HIF1α asparagine (N), pN-803, can be hydroxylated by FIH (see the Section 5.1) and inhibits the recruitment of HIF1α transcriptional coactivators p300/CBP, and, thus, HIF1α-dependent transcriptional activation [65]. In hypoxia, ubiquitination targeting of HIF1α pP-402, pP-564 does not take place because the hydroxylation of the HIF1α protein is absent. Von Hippel-Lindau factor (VHL) cannot recognize or attach ubiquitin to the newly synthesized HIF1α polypeptide, which results in increased HIF1α protein half-life. Consequently, HIF1α proteins start to accumulate in the nucleus and heterodimerize with aryl hydrocarbon receptor nuclear translocator (ARNT), resulting in activation of target genes (Figure 3) [66,67].

HIF1α’s importance for the hypoxic responses in the tissues of multicellular organisms cannot be overestimated. HIF1α regulates cell survival and differentiation, and also has proangiogenic functions. In the context of cell survival in hypoxia, however, very few HIF1α target genes were identified as essential in a genome-wide CRISPR knock-out screen of cell fitness in monocultures [32]. This would indicate that protein stabilization, as a concept of hypoxic responses of HIF1α, has evolved for purposes other than mere cell survival and might be an adaptation to hypoxia mostly used by multicellular organisms for tissue survival through blood flow restoration. Stabilization of the HIF1α protein can be detected after 15 min in pericytes, and peaks at 12 h after hypoxic insult [22,34]. Early activation of target genes can be detected after 2 h both in hypoxia and in oxygen and glucose deprivation in in vitro studies of pericytes. While HIF1α seems to be transcribing genes already after 2 h of hypoxia or OGD, a more numerous gene activation is seen at 6 h after hypoxia or OGD [34] in pericytes.

## 6. Regulator of G-Protein Signaling 5 (RGS5)

One of the first adaptive responses of pericytes to hypoxic environments is the expression of regulator of G-protein signaling 5 (RGS5). RGS5 is a negative regulator of G-protein-coupled receptors (GPCR) and, in the brain, RGS5 is exclusively expressed by pericytes [68].

RGS proteins work by regulating the time of active GPCR signaling. Once an agonist binds, the GPCR is activated by the binding of GTP to the G-protein α subunit (Gα), which then separates from the βγ dimer (Gβγ), enabling autonomous signaling from both subunits. RGS5 belongs to the R4/B subfamily—the smallest of RGS proteins—and acts as a GTPase activating protein, facilitating the intrinsic GTPase activity of the Gα-subunit [68]. Although it has been shown that RGS5 has preferential binding to the Gαi/o and Gαq subunits, recent studies have shown interactions with other molecules and various proteins, confirming regulatory functions beyond silencing GPCR signaling.

We have previously observed that RGS5 is upregulated in pericytes in hypoxic conditions and seems to be associated with a migratory movement away from the endothelial cell layer, leaving the endothelial cells unprotected and resulting in BBB leakage, a phenomenon that is associated with worse stroke outcomes [12]. Interestingly, knockout of RGS5 in pericytes prevents their migration from the capillaries after stroke, resulting in significantly reduced BBB leakage after stroke [69] or vessel stabilization in tumors [70]. This indicates a crucial role of RGS5 in the response of pericytes to hypoxia, and a function that is possibly related to migration (Figure 4, left panel).

## 7. RGS5 Regulation in Pericytes in Hypoxia Is Independent of HIF-1α

Interestingly, we, and others, have recently identified that RGS5 is indeed an oxygen sensor, where increased levels of RGS5 protein are observed under hypoxic conditions [22,29,70,71,72]. Importantly, RGS5 protein levels are rapidly stabilized under hypoxic conditions in human brain pericytes, but the hypoxic accumulation is independent of HIF-1α transcriptional regulation [22]. Although a previous study using human umbilical vein endothelial cells overexpressing RGS5 suggests transcriptional regulation under hypoxia, no HIF-1α binding sites in the RGS5 gene have yet been identified [73]. Instead, there is substantial evidence supporting the fact that the hypoxic induction of RGS5 is regulated post-translationally via the N-degron pathway [71,74,75,76].

The ubiquitin-dependent N-degron pathway regulates the protein half-life related to the identity of the proteins N-terminal residue. RGS5 is one of the proteins that has been identified as a substrate for the N-degron pathway, requiring a cysteine residue at position 2 (C2) for proteasomal degradation. Under normoxic conditions, the C2 residue is oxidized, allowing arginyl transferase-1 (ATE1) conjugation of arginine to the C2 N-terminal, thereby facilitating ubiquitin protein ligation, which primes RGS5 for degradation [71,74,75]. This highly conserved proteolytic pathway is likely essential to ensure appropriate levels of RGS5, giving it a short half-life and low abundance in the presence of oxygen. Thus, it is likely that RGS5 exerts its main function in response to hypoxia, whereby the C2 cysteine oxidation is prevented and RGS5 is stabilized, leading to rapid intra-cellular accumulation. In fact, it has been postulated that the continuous degradation of RGS5 in normoxia is a mechanism to facilitate rapid responsiveness to environmental changes, such as hypoxia, where RGS5 accumulation would inhibit GPCR-signaling and limit the transduction of external stimuli [71].

Interestingly, inhibition of RGS5 degradation in ATE1^-/-^ mouse strains results in embryonic lethality, with impaired cardiovascular formation and frequent hemorrhagic alterations. The authors also showed impaired late angiogenesis in E9.5–E13.5, where sprouted vessels had premature termination [77]. Given that RGS5, along with RGS4 and RGS16, is one of the main substrates of the ATE1-dependent N-degron pathway, it is possible that RGS5 has a regulatory role in vascular remodeling and that the abnormal vascular development in ATE1^-/-^ mice is the consequence of RGS5 accumulation.

## 8. Molecular Targets of RGS5 in Pericytes

### 8.1. G-Proteins

The biochemical properties of RGS5 have previously been characterized and shown to act as a GTPase-activating protein, interacting with Gα_i/o_ and Gα_q_ but not Gα_s_ nor Gα_12/13_ [78]. Thus, RGS5 plays a role as an effector in the cAMP-dependent pathway regulating adenylyl cyclase activity (Gα_i/o_) but also acts as antagonist to (Gα_q_)_,_ inhibiting phospholipase-Cβ, leading to a subsequent reduction in mitogen-activated protein kinase (MAPK) signaling [78,79]. As such, RGS5 has been shown to regulate signaling of several GPCR agonists known to exert vascular control, such as sphingosine-1 phosphate, angiotensin II and endothelin I [22,78,80,81]. In addition, a previous study on the function of smooth muscle cells (SMCs) during vascular remodeling showed the first evidence of RGS5 shifting G-protein signaling to Gα_12/13_-mediated Ras homolog family member A (RhoA) activation as the preferential signaling pathway due to RGS5 inhibition of Gα_q_ [82]_._

### 8.2. PDGFRβ

Endothelial cells secrete PDGF-BB, which binds to PDGF receptorβ (PDGFRβ) expressed on pericytes and leads to pericyte recruitment and vessel stabilization [25] (Figure 2). Indeed, PDGFBB stimulation has been shown to lead to rapid and sustained transcriptional down-regulation of RGS5 in SMCs and, when RGS5 is silenced in SMCs, the authors demonstrated that chemokinetic migration elicited by PDGFBB is induced [81].

We have recently been able to demonstrate that RGS5 desensitizes pericytes to PDGFBB-induced chemotaxis and high levels of RGS5 in hypoxic pericytes are associated with reduced phosphorylation of the PDGFRβ [22]. This inhibits pericyte recruitment and retention to the vascular wall and may also contribute to pericyte detachment from the blood vessels upon hypoxia, as RGS5 expression enables pericyte detachment, allowing angiogenic sprouting by endothelial cells (see Figure 3).

### 8.3. TGFβ—Smad2/3 Signaling

Data from pericytes in the tumor environment show that RGS5 upregulation in pericytes promotes apoptotic cell death due to the inhibition of Gα_i/q._ This leads to reduced phosphorylation of phosphoinositide 3-kinase (PI3K) and protein kinase B (AKT), resulting in an induced activation of caspase-9 and caspase-3 in the tumor microenvironment. Interestingly, tumors enriched with transforming growth factor-β (TGFβ) inhibited RGS5 binding to Gα_i/q_, thereby restoring pericyte survival. Furthermore, the authors showed direct binding of RGS5 to Smad2/3 or phopho-Smad2/3, thereby inhibiting the formation of the Smad transcription complex, which subsequently blocked its downstream genetic transcription activity [83].

## 9. Other Molecular Targets of RGS5

### 9.1. Wnt/β-Catenin

The Wnt signaling pathway is known to be essential in embryonic development, angiogenesis and differentiation [84]. A previous study showed that ATE1 has a regulatory role in Wnt signaling via inhibition of the transcriptional activity of β-catenin, although no support or evidence of β-catenin having an arginylation site has previously been shown [76]. Instead, the authors showed that RGS5 acts as the key intermediate mediator of ATE1 as the accumulation of intracellular RGS5 inhibits the phosphorylation of glycogen synthase kinase-3 beta (GSK3-β), and thereby blocks the degradation of WNT pathway transcriptional inducer and cofactor β-catenin. The authors also displayed that low ATE1 levels, which translates to accumulating levels of RGS5, predict poor prognosis in hepatic cellular carcinoma [76].

### 9.2. TAK1

Beyond its role as a modulator of GPCR signaling, a recent study showed that RGS5 directly binds to the transforming growth factor beta-activated kinase 1 (TAK1), a pivotal member of the MAPK signaling cascade [85]. Analogous to the inhibition of GSK3-β phosphorylation in Wnt signaling, RGS5 directly inhibits TAK1 phosphorylation and subsequent c-Jun-N-terminal kinase (JNK)/p38 activation. In this study, RGS5 was shown to have a protective role and reversed high-fat-diet-induced lipid accumulation in non-alcoholic fatty liver disease [85].

## 10. Outlook

The modification of the cellular and molecular response of pericytes to conditions such as hypoxia may allow therapeutic interventions that lead to more balanced adaption to the low oxygen levels. This could imply modulation of pericyte survival, migration and improved BBB integrity, which can result in reduced oedema and less hemorrhagic events after ischemic stroke. Furthermore, modulation of the inflammatory cascade elicited in pericytes after hypoxia may be important for the homeostasis of the brain microenvironment. Modulation of pericyte-specific responses to hypoxia may prove to be worthwhile by providing pericyte cell survival through metabolic adaptation to hypoxia, but also with regard to specific responses at the BBB interface. As such, RGS5 is needed for normal embryonic vessel and BBB formation, whereas in adult stroke pathology, RGS5 expression and hypoxic RGS5 protein stabilization leads to adverse stroke outcome [22,29,69,72]. This is likely contributed by non-responsiveness of pericytes to the PDGFBB secretion of nearby endothelial cells, causing the pericytes to detach from the vessel wall, disrupting tight junction and BBB integrity [22,69]. In such a scenario, blocking RGS5 temporarily in acute stroke appears to be a plausible therapeutic possibility. Another possibility is to trigger a transcription factor, such as STAT3 activity by cytokine, e.g., IL6 delivery in the acute phase of a stroke to trigger a pericyte response prior to endogenous IL6 release and response [86]. Further research will determine if regulation of pericyte hypoxic responses is a therapeutically viable option for treatment of conditions such as stroke.

## Figures and Tables

**Figure 1 ijms-24-05671-f001:**
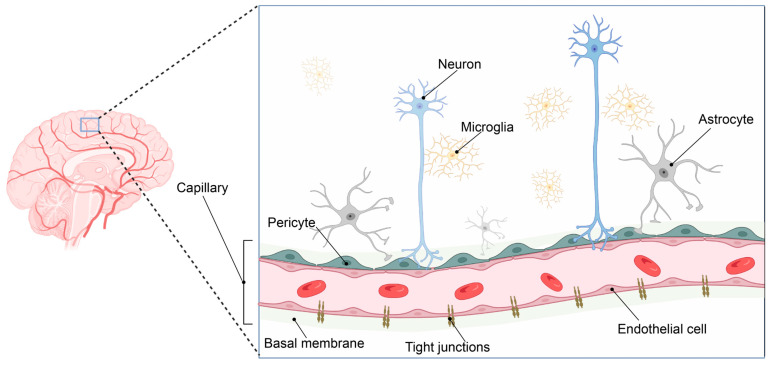
The neurovascular unit. The brain has evolved a vast vascular network to sustain its high oxygen and metabolic demands (left panel). The pial arterial vasculature of the brain branches into the microvasculature of the central nervous system, forming the neurovascular unit. This complex structure consists of an endothelial cell layer tightly connected via tight junction proteins. In addition, the endothelial cell layer, together with pericytes and astrocytic end-feet processes, is surrounded by the basement membrane, consisting of several extracellular matrix components. Neuronal synapse connections and local microglial cells complete the neurovascular unit, where close cellular interactions between the different cell types are essential to maintain brain homeostasis. Image created in BioRender.com, 18 February 2023.

**Figure 2 ijms-24-05671-f002:**
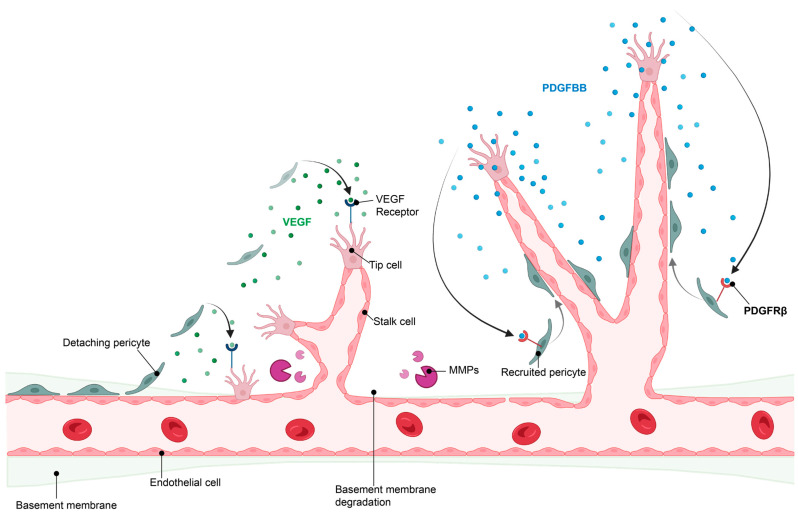
Hypoxia-induced angiogenesis. For angiogenesis to take place, pericyte detachment and migration is required, along with release of angiogenic factors such as VEGF and degradation of the basement membrane by matrix metalloproteinases (MMPs) to allow endothelial sprouting. Once nascent vessels are formed, pericyte recruitment is initiated by endothelial release of PDGFBB, which binds to the PDGFRβ on pericytes, stimulating pericyte recruitment and retention to the vascular wall, resulting in vessel stabilization. Image created in BioRender.com, 18 February 2023.

**Figure 3 ijms-24-05671-f003:**
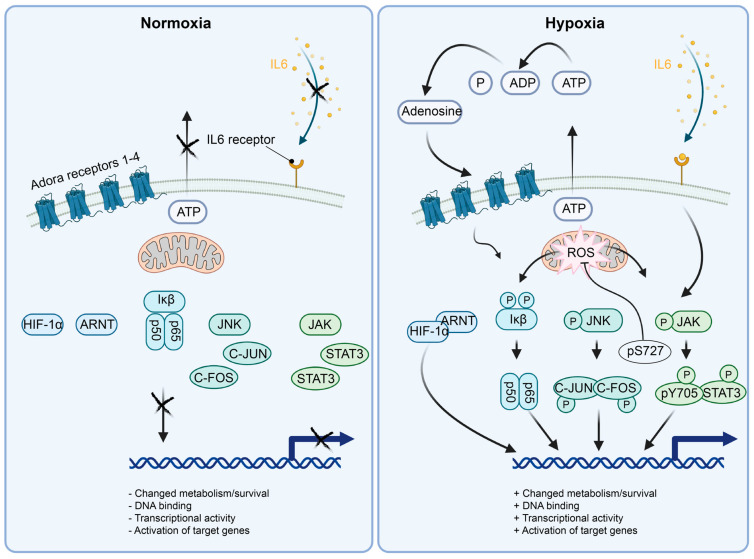
Schematics of the molecular mechanisms activating the selected transcription factors active in hypoxic pericytes. (Left panel) In normoxia, HIF1α is degraded by the presence of oxygen. NFκB (p50/p65) is kept inactive by the interaction with Iκβ. JNK is unphosphorylated by lack of active signaling, rendering C-JUN and C-FOS inactive. STAT3 is cytoplasmic and inactive in monomers due to the lack of JAK phosphorylation by absent upstream signaling. (Right panel) In hypoxia, HIF1α is stabilized by the absence of oxygen and translocated to the nucleus. NFκB (p50/p65) is activated and translocated to the nucleus by the phosphorylation of Iκβ, e.g., in a ROS-dependent manner. JNK is phosphorylated, for example, by mitochondrial ROS release followed by JNK activation of C-JUN and C-FOS, leading to AP1 (C-JUN/C-FOS heterodimer) formation and localization to the nucleus for transcriptional activity. Initial activation of STAT3 at p-S727 localizes the STAT3 protein to the mitochondrial lumen, where it affects the complex I, II and V, dampening ROS production. Subsequently, transcriptionally active STAT3 dimers are formed and localized to the nucleus, inducing transcriptional activity.

**Figure 4 ijms-24-05671-f004:**
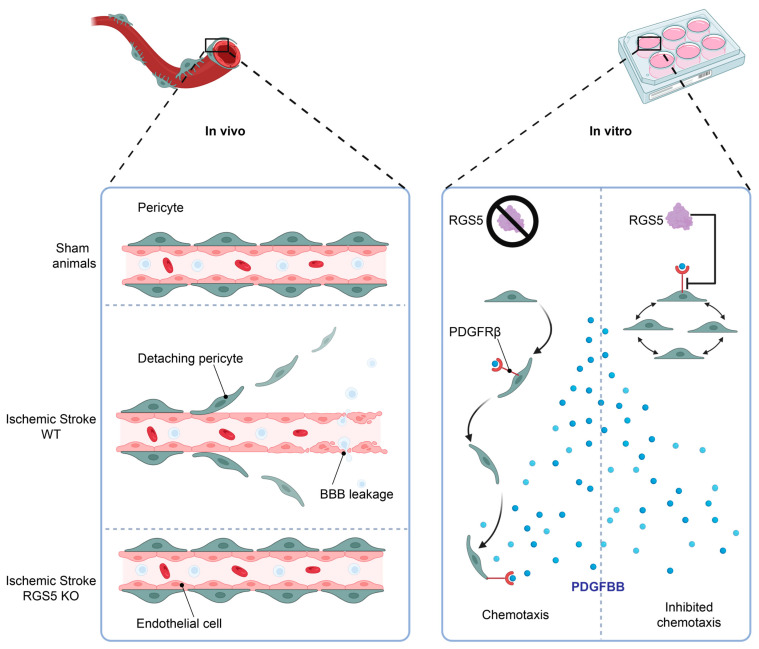
RGS5 regulates pericyte migration. Following stroke onset, the ischemic microenvironment induces RGS5 stabilization in WT pericytes, correlating to pericyte detachment from the blood vessels and migration into the brain parenchyma, leading to BBB leakage. However, in RGS5 KO mice, pericytes remain attached to the vascular wall, reducing BBB leakage and vascular damage (left panel). In addition, RGS5 expression in human brain pericytes inhibits PDGFBB chemotaxis (right panel), which could be a possible contributing factor to the pericyte detachment observed in vivo, as PDGFBB is a key regulator of pericyte recruitment and retention to the vascular wall. Created with BioRender.com, 18 February 2023.

## Data Availability

Not applicable.

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
