# Peer review of "Molecular Regulation of the Response of Brain Pericytes to Hypoxia"

_ijms, 2023, doi:10.3390/ijms24065671_

Round 1

Author Response

Reviewer 1:

" Molecular regulation of the response of brain pericytes to hypoxia"

Authors: Robert Carlsson, Andreas Enström1 and Gesine Paul

This is an interesting review of several mechanisms that may be involved in the brain pericytes

response to hypoxic conditions. The aims of the review are clear and the title is adequate and translates

the main focus of the review.

My main comment concerns the authors summary at the end of the manuscript called “Outlook”. In

my opinion this part should be a reflection and clear interpretation of the knowledge described and

its translation to new therapeutic interventions. Identification of the most relevant targets that may be

more promising for the modification of the cellular and molecular response of pericytes to hypoxia.

Best regards

We thank the reviewer for this positive review. We have added the targets that, based on the literature, hold the highest potential to be translated into therapeutic approaches.

Reviewer 2 Report

The Manuscript deal with an interesting topic. It is well organized, linear and clear and describe well the literature. The English have to be revised for some grammar and spelling errors.

Author Response

Reviewer 2:

The Manuscript deal with an interesting topic. It is well organized, linear and clear and describe well the literature. The English have to be revised for some grammar and spelling errors.

We thank the reviewer for the positive feedback. We have improved the grammar and spelling of the manuscript.

Reviewer 3 Report

The mansucript by Carlsson et al. with a title “Molecular regulation of the response of brain pericytes to hypoxia” gives a comprehensive overview of hypoxia-mediated changes in brain pericytes. Brain pericytes are part of the neurovascular unit and have distinct roles in maintaining blood-brain barrier function. Especially in pathological situations such as lack of oxygen, pericytes seem to play a central role and can serve as therapeutic target cells. Therefore, the studies investigating hypoxia-induced signaling in pericytes are important for the development of future therapeutic approaches. The review focusses on the role of the Regulator of G-protein Signaling 5 (RGS5) in pericytes and is therefore of interest to a specific readership. However, in addition to RGS5, the authors discuss common signaling pathways activated during hypoxia in the brain and brain pericytes, discuss the mechanisms of hypoxia-induced angiogenesis, signaling pathways and transcription factors that play a role in brain hypoxia. All in all, this is a valuable review that fits well with the scope of the journal.

I have only some comments:

Figure 1: it is not clear from the image that the ratio of pericytes to endothelial cells in the brain is 1:1. Shouldn’t all endothelial cells be completely covered by pericytes as shown in Figure 3?

I would change the order of section 8 and describe “transcriptional responses to hypoxia in pericytes” first, followed be the information on RGS5 (sections 5, 6 and 7).

In “Outlook” it would be good to briefly mention the RGS5 as a potential future therapeutic target.

Minor:

Line 136 : “the time of active(GPCR-signaling.- the parenthesis should be deleted

Author Response

Reviewer 3:

The mansucript by Carlsson et al. with a title “Molecular regulation of the response of brain pericytes to hypoxia” gives a comprehensive overview of hypoxia-mediated changes in brain pericytes. Brain pericytes are part of the neurovascular unit and have distinct roles in maintaining blood-brain barrier function. Especially in pathological situations such as lack of oxygen, pericytes seem to play a central role and can serve as therapeutic target cells. Therefore, the studies investigating hypoxia-induced signaling in pericytes are important for the development of future therapeutic approaches. The review focusses on the role of the Regulator of G-protein Signaling 5 (RGS5) in pericytes and is therefore of interest to a specific readership. However, in addition to RGS5, the authors discuss common signaling pathways activated during hypoxia in the brain and brain pericytes, discuss the mechanisms of hypoxia-induced angiogenesis, signaling pathways and transcription factors that play a role in brain hypoxia. All in all, this is a valuable review that fits well with the scope of the journal.

I have only some comments:

Figure 1: it is not clear from the image that the ratio of pericytes to endothelial cells in the brain is 1:1. Shouldn’t all endothelial cells be completely covered by pericytes as shown in Figure 3?

Response:

We have added a higher ration of pericytes to Figure 1. Specifically in the brain, the ratio between pericytes and endothelial cells is 1:1 to 1:3, and pericytes extend their processes to cover capillaries. We made this more clear in the image.

I would change the order of section 8 and describe “transcriptional responses to hypoxia in pericytes” first, followed be the information on RGS5 (sections 5, 6 and 7).

We have changed the order as suggested.

In “Outlook” it would be good to briefly mention the RGS5 as a potential future therapeutic target. 

We have included RGS5 as a potential future target to modulate the response of pericytes in hypoxia in the outlook section.

Minor:

Line 136 : “the time of active(GPCR-signaling.- the parenthesis should be deleted

Now line 426, removed.